# Phytochemical Characterization of By-Products of Habanero Pepper Grown in Two Different Types of Soils from Yucatán, Mexico

**DOI:** 10.3390/plants10040779

**Published:** 2021-04-15

**Authors:** Lilian Dolores Chel-Guerrero, Julio Enrique Oney-Montalvo, Ingrid Mayanín Rodríguez-Buenfil

**Affiliations:** Centro de Investigación y Asistencia en Tecnología y Diseño del Estado de Jalisco A.C., Subsede Sureste, Tablaje Catastral 31264, Km. 5.5 Carretera Sierra Papacal-Chuburn Puerto, Parque Científico Tecnológico de Yucatán, Mérida 97302, Yucatán, Mexico; lchelg_al@ciatej.edu.mx (L.D.C.-G.); juoney_al@ciatej.edu.mx (J.E.O.-M.)

**Keywords:** habanero pepper by-products, polyphenols, carotenoids, vitamins, soils

## Abstract

By-products of edible plants may contain potentially useful phytochemicals. Herein, we valorized the by-products of *Capsicum chinense* by phytochemical characterization of its leaves, peduncles and stems. Plants of habanero pepper were grown in a greenhouse, in polyethylene bags with two soils that were named according to the Maya classification as: K’ankab lu’um (red soil) and Box lu’um (black soil). Habanero pepper by-products were dried using an oven, the extracts were obtained by Ultrasound Assisted Extraction, and phytochemical quantification in all the extracts was conducted by Ultra Performance Liquid Chromatography coupled to Diode Array Detector (UPLC-DAD). Differences in the phytochemical content were observed according to the by-product and soil used. Catechin and rutin showed the highest concentrations in the peduncles of plants grown in both soils. The leaves of plants grown in black soil were rich in myricetin, β-carotene, and vitamin E, and the stems showed the highest protocatechuic acid content. While the leaves of plants grown in red soil were rich in myricetin and vitamin C, the stems showed the highest chlorogenic acid content. This novel information regarding the phytochemical composition of the by-products of *C. chinense* may be relevant in supporting their potential application in food and pharmaceutical industries.

## 1. Introduction

The search for novel healthy food sources and therapeutic agents is becoming more relevant due to the need to guarantee food security and the availability of safe, effective, and quality medicines for the growing world population [1,2,3]. Agricultural wastes are non-product outputs of production, and the processing of agricultural products that may contain material that can benefit man, but whose economic values are less than the cost of collection, transportation, and processing for beneficial use [4]. The term “by-product” suggests that plant food wastes might be usable and have their own market [5], and it is used to underline their large potentialities to be recycled as valuable products for industry. By-products are mainly made of skins, seeds, stems, leaves, wastewaters, and unusable pulp [6]. In this context, edible plants and their agricultural and food processing by-products are of great interest as important sources of nutrients, such as vitamins, proteins, carbohydrates, minerals, fibers, and phytochemicals, such as polyphenols, carotenoids, and terpenoids. In addition, phytochemicals have shown antiviral, anticancer, anti-inflammatory, and antimicrobial properties. Thus, the by-products can be used for the extraction of valuable components and/or novel pharmaceuticals, food supplements, or functional foods [7,8]. Particularly, in the food industry, these by-products can be used to fortify food products. Tumbas-Šaponjac et al. [9] incorporated anthocyanins and polyphenols, which were extracted from cherry pomace to cookies. The fortified cookies were sensorially acceptable. In another study, Carrullo et al. [10], reported that the enrichment of a cow milk kefir with the phenolic extract obtained by ultrasound assisted extraction (UAE) of skin of *Vitis vinifera* cv. Sangiovese, enhanced the total phenolic content, increased its total antioxidant capacity, and enhanced its inhibitory activity against α-amylase, α-glucosidase, and pancreatic lipase. The mechanisms of action of these phytocompounds in the human body are not clear. However, some studies have suggested that their activity in the human body may be due to antioxidant effects, inhibition of enzymes, inhibition of receptor activities, or stimulation or inhibition of genes expression [11].

Despite the various advantages, only 1 to 5% of all plants in the world have been scientifically studied. Mexico is a megadiverse country, with around 25,000 to 30,000 species of plants, of which 2,168 are edible [12]. One of these species is *Capsicum chinense* Jacq, commonly known as habanero chili. It belongs to the Solanaceae family, has been recognized with an Appellation of Origin, and it is of great commercial importance for the Yucatan Peninsula in the southeast region of Mexico. In 2019, this region had a production of 6,287.70 T of habanero chili in an area of 441.29 ha [13]. *C. chinense* is mainly grown in two types of soils that are characteristic of the Yucatán region, “Box luúm”, or black soils, and “K’ánkab lu’um”, or red soils [14], both being leptosols. The black soils have a higher content of calcium carbonates, organic matter, nitrogen, and phosphorus than red soils [15].

Chili peppers grow as a perennial shrub, but their lifespan depends on the climatic and cultivation conditions. The plants usually live for a decade or more, but it is mostly cultivated as an annual. This is the case of the *C. chinense* J. variety Jaguar with a harvest period of three to seven months in open field, and more than two years in protected conditions. In the Peninsula of Yucatan, 93% of habanero peppers are cultivated under an open field [13,16,17].

Nowadays, leaves and stems of Habanero pepper plants are discarded as waste and are usually burned after the peppers are harvested, whereas, during the industrialization, the peduncles are also discarded as waste; all are considered by-products. When considering the production of *C. chinense* of 6,287.70 T (22,000 plants ha^−1^ on average), and that 20% is industrialized, approximately 9.7 million plants (stems and leaves) and 237 million peduncles per year have been discarded as waste [13,18].

Leaves and fruits of *C. chinense*, have both been used in traditional medicine to treat asthma, flu, sore throat, gastrointestinal diseases, and arthritis [19]. The nutritive and bioactive potential of the fruit has been extensively studied. Studies have reported that the fruits have a high content of vitamin C and E and bioactive compounds such as polyphenols, carotenoids, and capsaicinoids. The fruits exhibit antioxidant, anticancer, and anti-inflammatory activities, mainly due to their capsaicin content [20,21]. However, there are few scientific studies on the nutritional and bioactive potential of its by-products.

Gayathri et al. [22] reported that the acetone:acetonitrile extract of the leaves of habanero chili grown in India contained polyphenols, alkaloids, tannins, flavonoids, and terpenoids. A study that was conducted by Herrera-Pool et al. [23] showed that the *C. chinense* variety Chichen Itzá (a hybrid of habanero pepper grown in greenhouse in the municipality of Suma, Yucatán, Mexico) contained: (1) phenolic acids and flavonoids in the leaves, which provide protection against oxidative stress and (2) phenolic acids in roots, which provide mechanical resistance and protection against pathogenic microorganisms. Another study investigated the methanolic extracts of the by-products of *C. chinense* variety Jaguar of plants that were grown in the soils of the state of Yucatán, and reported that the peduncles, leaves, and stems of plants that were grown in black soil could be used as potential sources of food. Specifically, the peduncles contained minerals, leaves contained protein and fat, and stems contained protein and fiber. The extracts (obtained by UAE) of the peduncles and stems of habanero pepper plants grown in red soil demonstrated potential for pharmaceutical application, due to the high antioxidant activity demonstrated by the DPPH (2,2-diphenyl-1-picrylhydrazyl) method. All of the metanolic extracts that were obtained by maceration and analyzed by colorimetric tests showed the presence of coumarins, polyphenols, and terpenoids, while saponins were only detected in the leaves [20]. A recent study on the hydromethanolic leaf extracts of *C. chinense* plants grown in Nigeria, evidenced there *in vitro* anti-inflammatory activity [24]. It is valuable to continue the evaluation and characterization of these plant materials due to these encouraging results. 

Taking the potential anti-inflammatory and antioxidant activity of the compounds present in the by-products of *C. chinense* and that no study has been reported on the characterization of the polyphenolic, carotenoid, vitamin, and capsaicinoid content in these samples into account, this study aimed to valorize the by-products of *C. chinense* Jacq. by phytochemical characterization of the leaves, peduncles and stems of the plants grown in two different types of soil from Yucatán. The results from this study could give support for the potential use of these by-products to develop novel products with food and pharmaceutical applications, and it could lead benefits to the environment, producers, industries, and consumers.

## 2. Results and Discussion

### 2.1. Polyphenol Content

Polyphenols are compounds of significant importance, because they can be used as additives in functional foods and exhibit health-promoting potential, especially due to their antioxidant, antibacterial, anticancer, immune system-promoting, and anti-inflammatory effects [25]. Polyphenols are unevenly distributed in different parts of the plant. They are associated with the growth, regulation, and plant structure (root system, stem, leaves, and reproductive structures) and they act as a defensive mechanism against aggression by herbivores and pathogens [26].

Overall, the phytochemical composition of the analyzed samples were similar. However, there were statistically significant differences in the content of the compounds detected in the samples (Table 1). Leaves and peduncles of plants grown in black soil exhibited the highest concentration of total polyphenols (154.04 ± 0.23 and 140.26 ± 0.46 mg 100 g^−1^ of dry basis, respectively). The higher concentration of polyphenols in leaves (Figure A2) and peduncles of plants that were grown in black soil can be associated with the physicochemical composition of this soil, which is characterized by being rich in nitrogen and organic matter [27]. These characteristics have shown an increase in the activity of the enzyme phenylalanine ammonia-lyasa (PAL), which has an important role in the biosynthesis of polyphenols in plants of the genus *capsicum* [28]. In the peduncles of plants that were grown in black soil (PBS) and red soil (PRS), catechin presented the highest concentration (47.11 ± 0.33 and 26.13 ± 0.16 mg 100 g^−1^ of dry basis, respectively) and the highest level of rutin was found in PBS (31.88 ± 3.90 mg 100 g^−1^ of dry basis). The leaves of plants grown in red soil (LRS) and black soil (LBS) contained the highest levels of myricetin, with a concentration of 32.04 ± 0.61 and 29.76 ± 0.68 mg 100 g^−1^ of dry basis, respectively, and hesperidin plus diosmin (21.27 ± 0.26 mg and 19.69 ± 0.42 100 g^-1^ of dry basis, respectively). In the stems of plants that were grown in black soil (SBS), the polyphenol with the highest concentration was protocatechuic acid (19.20 ± 0.18 mg 100 g^−1^ of dry basis). In the stems of plants grown in red soil (SRS), chlorogenic acid was found in the highest concentration (35.63 ± 0.20 mg 100 g^−1^ of dry basis).

Gallic acid was only detected in LBS (0.57 ± 0.01 mg 100 g^−1^ of dry basis) and coumaric acid, p-coumaric acid, cinnamic acid, vanillin, apigenin, diosmetin, kaempferol, quercetin + luteolin, neohesperidin, and naringenin were detected in all samples at a concentration range from 0.16 to 13.43 mg 100 g^−1^ of dry basis. These compounds are associated with the pathway of phenylpropanoids and flavonoids and they are characteristic of the genus *Capsicum* [29].

These findings are similar to those that were reported by Rodríguez-Buenfil et al. [20]. The authors harvested green and mature fruits of *C. chinense* grown in black, red, and brown soils of Yucatán at 142 days post-transplant (DPT) and then dried them in an oven by gravity convection at 65 °C for 72 h. Catechin was the polyphenol with the highest concentration, and its content was higher in the fruits of *C. chinense* grown in red soils (61.64 ± 7.55 mg 100 g^−1^ of dry basis) than in the by-product samples that were analyzed in this study. Nevertheless, the content of rutin and myricetin were higher in the by-products analyzed than those reported in these fruits (29.14 ± 6.33 mg 100 g^−1^ of dry basis, 6.76 ± 2.06 mg 100 g^−1^ of dry basis). Similarly, Nagy et al. [30] reported that the content of myricetin was the highest in the by-products than in the fruits of *C. frutescens* (2.57 ± 0.18 mg 100 g^−1^ of dry basis). 

The total content of polyphenols in the by-products of *C. chinense* was in the range of 56.94 ± 0.16 and 154.04 ± 0.23 mg 100 g^−1^ of dry basis, being comparable to the polyphenols and anthocyanin content of sour cherry pomace extract (78.99 mg of gallic acid equivalents per 100 g of dry basis) encapsulated in whey that was incorporated in cookies and increased their functional characteristics [9]. Subsequently, the quantity of polyphenols in the *C. chinense* by-products analyzed could be enough to use as a food ingredient with similar results.

The difference in the levels of polyphenols that was detected in the analyzed samples could be due to the association between the function of the compounds in the plant’s life cycle and vegetative phase. It may also be related to several other factors that can influence this behavior, such as the climate, time of year, type of soil, type of plant organ, and, in this case, the temperature, and time conditions of the extraction [15,20,31,32].

### 2.2. Carotenoid Content

Carotenoids are natural pigments that are distributed in photosynthetic bacteria, some species of archaea and fungi, algae, plants, and animals. There are two types of carotenes: without oxygen in their terminal rings (for example, β-carotene and lycopene) and xanthophylls, which contain oxygen in their terminal rings (for example, lutein). Carotenoids play a vital role in plants, acting in cellular antioxidation and gene regulation. Furthermore, these compounds represent a good alternative for the pharmaceutical and food industries, especially as health foods, as they exhibit antioxidant and provitamin A activities. In addition, these compounds have also been reported to prevent diseases, such as cancer, macular degradation, and cataract [33].

β-carotene was identified in all of the analyzed samples (Figure 1a). Statistically significant differences were observed in the content of β-carotene in all the by-products (*p* ≤ 0.05), and the interaction of type of soil and by-product showed an effect on the concentration of β-carotene. The leaves of plants that were grown in black soils (Figure A3) showed the highest β-carotene content (161.10 ± 8.20 mg 100 g^−1^ of dry basis), while other samples presented β-carotene levels that ranged from 7.30 ± 0.14 to 118.98 ± 1.13 mg 100 g^−1^ of dry basis. In contrast, lutein was not detected in any of the samples analyzed.

β-carotene is a carotenoid that is specifically found in plants and fruits. Its biosynthesis is associated with the different stages of fruit maturity stages, in which carotenoids have diverse profiles and at varying levels [34]. The fruits of *C. chinense* collected in August 2018 and leaves of *C. annuum* were reported to contain β-carotene and lutein [20], and the by-products contained β-carotene, but not lutein. In this study, the β-carotene content in leaves and peduncles (LBS, LRS, PBS, and PBS) was higher than the reported in immature fruits of *C. chinense* grown in black soil of Yucatán, Mexico (10.78 ± 1.69 mg 100 g^−1^ dry basis). In the leaf tissue of *C. annuum* that was grown under three different illumination conditions, the highest content reported was 11.86 mg 100 g^−1^ fresh weight leaf [34].

Furthermore, the leaves of plants grown in black soil had similar contents to those of *Solanum macrocarpon* (β-carotene content = 140.05 ± 0.13 mg 100 g^−1^ dry basis), which has been reported to be a rich source of carotenoids [35]. Therefore, these leaves should also be considered to be a rich source of carotenoids.

The absence of lutein in the by-products of *C. chinense* could be attributed to the amount of light to which the crop was exposed, or the content of salt presented in the irrigation water, because these factors have an effect on the phytoene enzyme synthetase, which limits the speed of carotenoid biosynthesis, altering the production of carotenoids in the fruit or in the plant [36,37,38]. For example, it was reported that the partially invasive light protection of carotenoids leads to photodegradation, which results in the production of β-carotene-5,6-epoxide, together with lutein-5,6-epoxide in green leaves or in salt-stress situations (150 mM NaCl); therefore, the synthesis of quercetin 3-β-D-glucoside was produced by the reduced β-carotene and lutein [36,37,38].

### 2.3. Vitamin Content

Vitamins are one of the main nutrients that play an important role in the body. To ensure growth and development and maintenance of good health, vitamins must be supplied in the diet. Vitamins can be divided into fat-soluble vitamins (A, D, E, and K) and water-soluble vitamins (B-complex and C). Vitamin A helps in the maintenance of healthy mucous membranes, skin, vision, tooth, and bone growth, vitamin E helps to protect the cell walls, and vitamin C is involved in many important physiological processes, such as iron absorption and the immune response [39,40].

Figure 1b–d, present the results of the quantification of vitamins A, C, and E, respectively. Figure A4 show a chromatogram of leaves extracts that were obtained from habanero pepper plants grown in black soil to analyze vitamin A and E, while Figure A5 shows a chromatogram of leaves extracts obtained from habanero pepper plants grown in red soil to analyze vitamin C. The statistical analysis indicated significant differences among the samples. In addition, the interaction between the type of soil and by-product showed an effect on vitamin content. Vitamin A was detected in low concentrations in the leaves of plants grown in both types of soil and in the peduncles of plants that were grown in black soil, with concentrations ranging from 0.35 ± 0.02 to 0.84 ± 0.03 mg 100 g^−1^ of dry basis. Vitamin C and vitamin E were detected in all of the by-products. The leaves of plants grown in red soil (LRS) showed the highest content of vitamin C (16.71 ± 0.20 mg 100 g^−1^ of dry basis) and leaves of plants grown in black soil (LBS) presented the highest content of vitamin E (25.58 ± 0.02 mg 100 g^−1^). 

Rodríguez-Buenfil et al. [20] studied the vitamins of fruits of *C. chinense.* They did not detect vitamin A, and showed that the degree of maturity and type of soil had a significant effect on the concentration of vitamins C and E. The highest concentration of vitamin C was detected in chili plants that were grown in black soil, 136.55 ± 0.36 mg 100 g^−1^ of dry basis (higher than that exhibited in the studied by-products), while the highest concentration of vitamin E was found in chili peppers that were harvested from plants that were grown in red soil, 9.27 ± 2.06 mg 100 g^−1^ of dry chili (being lower than the concentration presented by the peduncles and leaves of plants grown in both types of soil). Similarly, the interaction of type of soil and by-product also had a significant effect on the concentration of these vitamins in the by-products (Figure 1b–d).

These discrepancies between the content of vitamin C and vitamin E, in different organs (by-products) of *C. chinense* of plants that were cultivated in the same type of soil may be due to a greater accumulation of sugars in fruits (sink organs) than in leaves (source organs), which are precursors of vitamin C [41,42], differences in vitamin C and vitamin E content have also been observed in different organs of *Fragaria × ananassa*, Duch., *Vaccinium corymbosum* L., and *Capsicum annuum.* In addition, this process of nutrient translocation depends on the control of ecological factors, such as humidity, drought, light and temperature, and stress situations [41,43,44].

### 2.4. Capsaicinoid Content 

Capsaicinoids are a group of alkaloids responsible for the pungency of fruits of the genus *Capsicum*. Capsaicin and dihydrocapsaicin are the two most common capsaicinoids in pepper (90%). Capsaicinoids have remarkable antitumor properties; however, their clinical applications are still very limited due to their low selectivity and high toxicity (LD50 0.5 to 5 g kg^−1^) [45]. 

This study quantified the levels of capsaicin and dihydrocapsaicin in the peduncles, leaves, and stems of plants that were grown in black and red soil. These capsaicinoids were not detected in any by-products. This was consistent with the other studies that reported that capsaicin and dihydrocapsaicin are most abundant in the fruit, but absent in vegetative organs [46].

## 3. Materials and Methods

### 3.1. Chemical Reagents

HPLC grade solvents were used for the ultra-performance liquid chromatography (UPLC) analysis. Methanol (PubChem CID: 887), water (PubChem CID: 962), hexane (PubChem CID: 8058), acetonitrile (PubChem CID: 6342), acetic acid (PubChem CID: 176), and formic acid (PubChem CID: 284) were obtained from Sigma Aldrich (Steinheim, Germany).

The standards that were used for the quantitative analysis were obtained from Sigma Aldrich ®. The standards used and their respective purity were the following: β-carotene (Purity ≥ 95%), lutein (Purity ≥ 98%), α-tocopherol (Purity ≥ 96%), ascorbic acid (Purity ≥ 99%), capsaicin (Purity ≥ 95%, dihydrocapsaicin (Purity ≥ 95%), gallic acid (Purity ≥ 97%), protocatechuic acid (Purity ≥ 98%), chlorogenic acid (Purity ≥ 95%), coumaric acid (Purity ≥ 98%), p-coumaric acid (Purity ≥ 98%), cinnamic acid (Purity ≥ 98%), vanillin (Purity ≥ 99%), catechin (Purity ≥ 98%), myricetin (Purity ≥ 96%), apigenin (Purity ≥ 95%), diosmetin (Purity ≥ 99%), rutin (Purity ≥ 94%), kaempferol (Purity ≥ 90%), quer-cetin (Purity ≥ 95%), luteolin (Purity ≥ 98%), hesperidin (Purity ≥ 80%), diosmin (Purity ≥ 90%), neohesperidin (Purity ≥ 90%), and naringenin (Purity ≥ 95%). 

### 3.2. Samples and Sample Preparation

#### 3.2.1. Plant Material

The leaves, stems, and peduncles of *C. chinense* J., variety Jaguar (variety register number CHL-008-101109) were obtained by cultivating the plant in a greenhouse (15 plants for each type of soils), with controlled irrigation and fertilization conditions [15], at the Centro de Investigación y Asistencia en Tecnología y Diseño del Estado de Jalisco, A.C. (CIATEJ) Subsede Sureste (Latitude N 21º 8 ’1.288” and Longitude W 89º 46′52.26”). The crops were planted in polyethylene bags that were filled with 12 kg of two types of soil characteristic of the Yucatán region: red soil (*K’áankab lu’um*) and black soil (*Box lu’um*), which were obtained from a local supplier located in Mérida, Yucatán, Mexico). The plants were obtained on harvest number 12 (the last expected harvest, after which the plant is generally wasted), 265 days after the transplantation (DAT) of seedlings (45 days of growth with a minimum height of 19.3 cm and ten true leaves) from the Cutz nursery in Suma de Hidalgo, Yucatán, México, which is characterized by the use of certified seeds. The ambient temperature was in the range of 24 to 47 °C and the relative humidity was 91%.

#### 3.2.2. Drying of Habanero Pepper By-Products 

Dried habanero pepper by-products were obtained according to the methodology that was reported by Rodríguez-Buenfil et al. [20]. Briefly, different parts of the plants were separated into peduncles, leaves, and stems. The stems were chopped using a knife for easy handling. Subsequently, the separated parts were dried in a stainless-steel oven, model HS60-AID (Novatech, Jalisco, México). The peduncles and stems were dried to 44 °C for 48 h, and the leaves were dried at 44 °C for 240 h to reduce their moisture content to <5% for adequate handling and storage. Subsequently, the plant parts were grinded and sieved (pore size 500 μm, Sieve # 35, Fisher Scientific, Boston, MA, USA). The obtained powders were stored at −20 °C until analysis.

#### 3.2.3. Extraction of Polyphenols

The extraction of polyphenols was performed according to the methodology that was described by Oney-Montalvo et al. [15]. Briefly, a quantity of 0.5 g of each by-product of habanero pepper was mixed with 2.5 mL of solvent, which was composed of methanol:water (80:20), in 15 mL Falcon tubes. The mixture was sonicated at 42 kHz for 30 min. at 28 °C (Branson sonicator, model 3510, Danbury, EE.UU.) and then centrifuged at 4816× *g* and 4 °C for 30 min. (Centrifuge Mega Fuge 40 R, Thermo scientific, Langenselbold, Germany). The supernatant was collected, and the centrifugation and recovery of the supernatant was repeated. Finally, the samples were filtered through a 0.2 µm PTFE filter and they were deposited in amber vials for analysis by UPLC.

#### 3.2.4. Extraction of Carotenoids

The extraction of carotenoids was performed according to the methodology that was described by Rodríguez-Buenfil et al. [20]. Briefly, a quantity of 0.5 g of each by-product of habanero pepper was weighed and placed in 15 mL Falcon tubes, followed by the addition of 4 mL of hexane. The mixture was homogenized with the help of a vortex mixer. The tubes were sonicated for 20 min. at 42 kHz, and then the extract was centrifuged at 3500 rpm and 4 °C for 30 min., the supernatant was evaporated and resuspended in 1 mL of hexane. Finally, the samples were filtered through a nylon membrane with a pore size of 0.22 μm and then deposited in amber chromatographic vials for UPLC analysis.

#### 3.2.5. Extraction of Vitamins and Capsaicinoids

The extraction procedure was performed, as described by Rodríguez-Buenfil et al. [20]. Briefly, a quantity of 0.05 g of each by-product of habanero pepper was placed in a 15 mL Falcon tube, followed by the addition of 4 mL of water:acetonitrile (80:20). The mixture was homogenized with the help of a vortex mixer. Subsequently, the samples were sonicated for 20 min. at 42 kHz, and the extracts were filtered through a nylon membrane filter with a pore size of 0.2 μm and then deposited in amber chromatographic vials for UPLC analysis. 

### 3.3. Phytochemical Determination by UPLC-DAD

Phytochemical quantification was conducted using an UPLC Acquity H Class (Waters, Milford, MA, USA) with a diode array detector (DAD). The column was an Acquity UPLC HSS C18 (100 A°, 1.8 µm, 2.1 mm × 50 mm) (Waters, Milford, MA, USA).

#### 3.3.1. Determination of Polyphenol Content

The quantification of polyphenols by UPLC-DAD was done using the following conditions: flow speed of 0.5 mL min.-1 with a column temperature set at 45 °C and injection volume of 2 µL. The measurement was performed using the DAD at 280 nm. The polyphenols were separated using acetic acid (0.2%) as solvent A and acetonitrile with acetic acid (0.1%) as solvent B. The elution conditions were as follows: 0–10 min. from 1% B to 30% B; 10–12 min. 30% B; and, 12–15 min. from 30% B to 1% B.

The quantification of the polyphenols was done using two external calibration curves prepared with 17 standards of polyphenols from Sigma Aldrich^®^ (Figure A1). The first curve was conformed with: gallic acid, protocatechuic acid, chlorogenic acid, coumaric acid, cinnamic acid, catechin, rutin, kaempferol, quercetin, and luteolin. While, the second curve was conformed with: p-coumaric acid, vanillin, myricetin, apigenin diosmetin, hesperidin, diosmin, neohesperidin, and naringenin. In both cases, stock solutions (1 mg mL^−1^) of all standards were first prepared. Next, the calibration curves were prepared in the range of 1 to 75 µg mL^−1^. The polyphenols were identified in the samples by comparing their retention time with those of the standards; this method was described by Oney-Montalvo et al. [15]. Quercetin and luteolin, and diosmin and hesperidin were determined together because the peaks coeluted.

#### 3.3.2. Determination of Carotenoid Content

The chromatographic conditions for the analysis of carotenoids were those used by Rodríguez-Buenfil et al. [20]. Briefly, they consisted of an isocratic mobile phase, which was composed of acetonitrile:methanol (70:30). The flow rate was 0.5 ml min.^−1^. The column temperature was 35 °C, injection volume was 2 µL, and the wavelength was 475 nm. The quantification of carotenoids was performed using a calibration curve, prepared with β-carotene and lutein as the external standards, which were procured from Sigma Aldrich.

#### 3.3.3. Determination of Vitamin Content (A, C and E)

Vitamin A and E levels were determined using the method that was developed by Rodríguez-Buenfil et al. [20], which consists in a mobile phase of acetonitrile:methanol (50:50) with a flow speed of 0.5 mL min.^−1^, a column temperature set at 35 °C, and an injection volume of 2 μL. The measurement was performed using the DAD at 290 nm. Quantification was carried out by external calibration with α-tocopherol standards. The calibration curve was prepared in the range of 1.5 to 75 μg mL^−1^. The vitamins were identified in the samples by comparison of the retention time with that of the standard.

On the other hand, the vitamin C levels were determined using the method that was established by Rodríguez-Buenfil et al. [20], consisting in a flow speed of 0.25 mL min.^−1^ of water with formic acid at 0.1% in a column with the temperature set at 27 °C and an injection volume of 2 μL. The measurement was performed using the DAD at 244 nm. Quantification was carried out by external calibration with ascorbic acid in the range of 0.5 to 5 μg mL^−1^. Vitamin C was identified in the samples by comparison of the retention time with that of the standard.

#### 3.3.4. Determination of Capsaicinoid Content

The chromatographic conditions for the analysis of capsaicin and dihydrocapsaicin consisted of an isocratic mobile phase that was made up of acetonitrile (mobile phase A) and water with 0.1% formic acid (mobile phase B) in a ratio of 60:40%. The flow rate was 0.2 mL min.-1, the column temperature set was 27 °C, the injection volume was 2 μL, and the wave-length was 280 nm. The calibration curve was prepared in the range of 0.005 to 0.08 mg mL^−1^. The total capsaicinoids were reported as the sum of capsaicin and dihydrocapsaicin; Morozova et al. described this method [21].

### 3.4. Validation of the Analytical Methods

The validation of the analytical methods used in the present work was made according to the work that was carried out by Shrivastava et al. [47] and using as a reference the “Eurachem Guide: The Fitness for Purpose of Analytical Methods—A Laboratory Guide to Method Validation and Related Topics” [48]. (Table A1 and Table A2 in Appendix A).

### 3.5. Statistical Analysis

The results of the experiments were expressed as the mean ± standard deviation. Statistically significant differences between the groups were calculated using a two-way analysis of variance (ANOVA). The interactions between groups were analyzed using a multifactorial analysis of variance (the factors analyzed were soil type and by-product type and the interactions between them), followed by the Tukey test. A *p* ≤ 0.05 was considered to be statistically significant. Data analysis was performed using the statistical package Statgraphics Centurion 18-X64. For all analyses, determinations were made in triplicate as independent experiments, except for protocatechuic acid in leaves of plants that were grown in black soil (LBS), chlorogenic acid in leaves of plants grown in red soil (LRS) and stems of plants grown in black and red soil (SBS and SRS), coumaric acid in peduncles and stems of plants grown on red soil (PRS and SRS), ρ-coumaric acid in leaves of plants grown in black soil (LBS) and peduncles of plants grown in black soil and red soil (PBS and PRS), cinnamic acid (SRS), vanillin (LBS, PRS), catechin (SBS, LRS, SRS), myricetin (SRS), apigenin (LBS, PRS and SRS), diosmetin (SBS), rutin (SRS), kaempferol (PBS and SRS), hesperidin + diosmin (SRS), and neohesperidin (SRS), which were done in duplicate (in the absence of more sample).

## 4. Conclusions

The phytochemical characterization of the by-products of *C. chinense* variety Jaguar showed that all of the materials studied had similar compounds with varying levels, dependent on the soil used. Only in the cases of chlorogenic and coumaric acid, myricetin, quercetin plus luteolin, hesperidin plus diosmin, and Vitamin C, the by-products of plants grown in red soil exhibited a higher content than the by-products of plants that were grown in black soil. In the case of all other phenolic compounds, carotene, vitamin A, and vitamin E, the content was the highest in the by-products of plants grown in black soil. These by-products are readily available, and they are good sources of nutrients and bioactive compounds, rich in high quantities of polyphenols, carotenoids, and vitamins, mainly rutin, myricetin, β-carotene, and vitamin E. Thus, these by-products can be considered to be a potential source of those compounds or for the development of novel products to be used in the food or pharmaceutical industries. This, in turn, may help food security, climate change, and lack of access to costly drugs, three major issues of concern by the global population. To the best of our knowledge, this is the first study that characterized the polyphenol, carotenoid, vitamin, and capsaicinoid profile in *C. chinense* by-products. Further, the promotion of the integral use of these edible plants may contribute to the reduction of plant waste and may have a social and economic impact by increasing the profitability of the crop and its industrialization.

## Figures and Tables

**Figure 1 plants-10-00779-f001:**
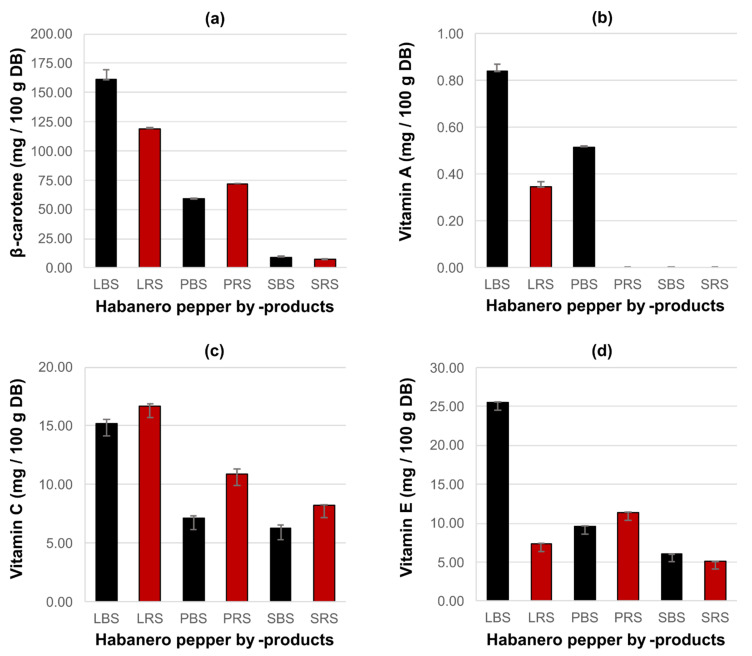
The effect of the factors type of soil and by-product on different response variables: (**a**) β-carotene content, (**b**) Vitamin A content, (**c**) Vitamin C content, and (**d**) Vitamin E content. The error bars on the figure represent the standard deviation (n = 2). DB: Dry basis; LBS: Leaves of plants grown in black soil; LRS: Leaves of plants grown in red soil; PBS: Peduncles of plants grown in black soil; PRS: Peduncles of plants grown in red soil; SBS: Stems of plants grown in black soil; SRS: Stems of plants grown in red soil.

**Table 1 plants-10-00779-t001:** Polyphenols content in chile habanero by-products extracts of plants grown in black and red soils of Yucatán, México.

Polyphenol ^A^	Black Soil			Red Soil		
	Leaves (LBS)	Peduncles (PBS)	Stems (SBS)	Leaves (LRS)	Peduncles (PRS)	Stems (SRS)
Gallic Acid	0.57 ± 0.01 ^a^	―	―	―	―	―
Protocatechuic Acid	0.00 ± 0.00 ^d*^	0.00 ± 0.00 ^d^	19.20 ± 0.18 ^a^	1.58 ± 0.08 ^b^	0.00 ± 0.00 ^d^	0.85 ± 0.03 ^c*^
Chlorogenic Acid	16.79 ± 0.21 ^b^	14.24 ± 0.80 ^c^	4.12 ± 0.02 ^e*^	2.92 ± 0.03 ^f*^	8.56 ± 0.24 ^d^	35.63 ± 0.20 ^a*^
Coumaric Acid	7.46 ± 0.06 ^b^	3.28 ± 0.05 ^c^	3.26 ± 0.02 ^c^	7.92 ± 0.54 ^a^	2.24 ± 0.02 ^d*^	0.79 ± 0.04 ^e*^
ρ-coumaric Acid	5.29 ± 0.03 ^a*^	1.93 ± 0.87 ^b*^	2.30 ± 0.01 ^b^	1.95 ± 0.04 ^b^	0.87 ± 0.00 ^c*^	0.65 ± 0.02 ^c^
Cinnamic Acid	1.67 ± 0.04 ^c^	4.01 ± 0.12 ^a^	1.88 ± 0.22 ^c^	1.76 ± 0.55 ^c^	2.77 ± 0.03 ^b^	2.65 ± 0.05 ^b*^
Vanillin	9.29 ± 0.01 ^a*^	9.53 ± 0.95 ^a^	3.74 ± 0.02 ^d^	7.06 ± 0.10 ^b^	5.30 ± 0.02 ^c*^	0.16 ± 0.00 ^e^
Catechin	3.36 ±0.20 ^c^	47.11 ± 0.33 ^a^	2.03 ± 0.09 ^d*^	2.96 ± 0.22 ^c*^	26.13 ± 0.16 ^b^	2.44 ± 0.18 ^d*^
Myricetin	29.76 ± 0.68 ^b^	6.11 ± 0.17 ^c^	1.12 ± 0.03 ^f^	32.04 ± 0.61 ^a^	3.31 ± 0.03 ^e^	4.97 ± 0.01 ^d*^
Apigenin	6.62 ± 0.52 ^a*^	1.33 ± 0.05 ^c^	1.19 ± 0.08 ^c^	1.72 ± 0.16 ^b^	0.64 ± 0.03 ^d*^	0.77 ± 0.08 ^d*^
Diosmetin	7.83 ± 0.65 ^a^	2.81 ± 0.16 ^c^	0.73 ± 0.02 ^d^	4.45 ± 0.38 ^b^	0.80 ± 0.07 ^d^	2.41 ± 0.12 ^c*^
Rutin	10.84 ± 0.27 ^c^	31.88 ± 3.90 ^a^	7.88 ± 1.85 ^c^	9.63 ± 1.43 ^c^	18.65 ± 0.09 ^b^	3.95 ± 0.03 ^d*^
Kaempferol	7.61 ± 0.13 ^a^	1.11 ± 0.01 ^c*^	1.07 ± 0.01 ^cd^	1.91 ± 0.04 ^b^	0.95 ± 0.01 ^d^	1.94 ± 0.14 ^b*^
Quercetin + Luteolin **	13.19 ± 0.12 ^a^	3.54 ± 0.16 ^b^	1.23 ± 0.01 ^d^	13.43 ± 0.46 ^a^	2.36 ± 0.01 ^c^	3.79 ± 0.85 ^b^
Hesperidin + Diosmin **	19.69 ± 0.42 ^b^	4.98 ± 0.06 ^c^	1.33 ± 0.04 ^e^	21.27 ± 0.26 ^a^	2.90 ± 0.02 ^d^	0.71 ± 0.09 ^f*^
Neohesperidin	11.79 ± 0.31 ^a^	7.46 ± 0.12 ^c^	5.38 ± 0.06 ^d^	11.25 ± 0.35 ^b^	4.36 ± 0.04 ^e^	1.88 ± 0.00 ^f*^
Naringenin	2.28 ± 0.18 ^a^	0.94 ± 0.05 ^c^	0.48 ± 0.10 ^d^	2.07 ± 0.15 ^b^	0.51 ± 0.02 ^d^	0.36 ± 0.03 ^d^
Total polyphenols quantified^B^	154.04 ± 0.23 ^a^	140.26 ± 0.46 ^b^	56.94 ± 0.16 ^e^	123.92 ± 0.32 ^c^	80.35 ± 0.05 ^d^	63.95 ± 0.11 ^e^

^A^ Expressed as mg 100 g^−1^ dry basis; mean ± SD (n = 3); * mean ± SD (n = 2); ** were determined together because separation of the peaks by UPLC was not clear; ^B^ Expressed as the sum of the individual polyphenols contents in the analyzed samples as mg 100 g^−1^ dry basis; ^a–f^ Different superscript letters in the same row indicates significant differences (*p* ≤ 0.05).

## Data Availability

Not applicable.

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
