# Peer review of "Phytochemical Characterization of By-Products of Habanero Pepper Grown in Two Different Types of Soils from Yucatán, Mexico"

_plants, 2021, doi:10.3390/plants10040779_

Round 1
Reviewer 1 Report
The authors performed an interesting study on the phytochemical composition of different plant parts (what they called by-products) of Capsicum chinense cultivated in two different soil types from Mexico. The study presents novelty, since it describes for the first time the phytochemical composition of plant parts different from the fruits (commonly studied) and also applying two different soil types. Few English and typing corrections are required (signed in yellow) but several other corrections are needed before the acceptence by Plants. The main aspects are presented bellow and some minor corrections are marked in the PDF.
- First of all, the authors call the plant parts as "by-products", assuming that they are" discard products" from the the harvesting of the chili fruits (peper production). They also present some numbers relative to the waste in this production (lines 67-72 of the introduction). However, being a perennial plant (not annual), these plant parts would not be considerer as by-products in a first moment, since with the carefuly maintanence of the plants, they should be able to produce new fruits in other frutification periods. I understand that some authors classify this species as "semi-perennial", considering the difficults to keep them, or that maybe the harvesting for the fruit utilization is made collecting the entire plants from the soil. However, these plant parts cannot be considered as by-products if we think the species as perennial. In order to avoid misunderstandings the authors could include in the paragraph in the introduction (l. 67-72) some sentences about these aspects. This would inform to the reader that even not being annual, these plants are not easyly kept over sequential seasons and/or that the harvesting process "kills" the entire plant;
- in the abstract the authors should clarify the methods to verify the effect of the types of soil (it is not clear in the abstract if they cultivated the plants or if they were harvested from natural places);
- in l.67 there are two references without being numbered;
- l.112-113 - this information should be presented in M&M and as a footnote of the table 1;
- l.114-116 - please re-phrase the sentence and put together with the first paragraph;
- L.122-123 and M&M (quantification of polyphenols): Why did the authors present the total phenolic content in gallic acid equivalents, if they used different standards (calibration curves) to quantify the different compounds?
- Please, regarding the quantification unit, use mg mL-1 instead of mg 100 g-1 (also, in some "-1" there are two "minus signs');
- Why didn't the authors conduct and present the results of a two-way anova instead if an one-way? (they have two independet factors: type of soil and plant "by-product");
- The sentences in lines 168 and 169 are "out of place", being "lost" in the text and the sentence l171-177 is too long and should be rephrased;
- The discussion about the polyphenolic characterization and their comparison is vogue. The authors should better develop the idea and discuss more about:
- the different classes/ compounds individually (their different roles in different plant parts and according to the soil characteristics);
- the effects of soil properties in the content of these different compounds (which could be the characteristics or composition of the soils that could explain the differences that they found?). - In Fig.1 the authors should provide a better quality figure.
Besides, as the "x axis" correspond to nominal variable (not continous), the authors could present the data as bars, not as lines. - The sentences in l.259-262 are repetitive and have been already described above (l.247-250).
- The discussion in l.270-271 should be "re-thought". The authors are comparing the content in leaves (that they in this study found) with the content in fruits, found in a previous study. The previous study used the same conditions (growing, extraction and analysis) than those used in the present study? Is this comparison really meaningful?
- L. 305 - please specify the origin of the soils
- l.421-428 - why did the authors use duplicate for these parameters instead of triplicate, while for other polyphenols they have the triplicate? Didn't they use the same extracts to quantify all polyphenols?

Author Response
"Please see the attachment."

Reviewer 2 Report
The present manuscript represents an interesting characterization of bioactive compounds in different by-products of Habanero pepper, by considering also two types of soils, which can affect the compositions of the compounds, object of this study.
The characterization of nutraceuticals from by-products is nowadays a topic of great interest, due to the positive impact that it may represent from the economic and environmental point of view.
The introduction well explains this concept with a description of the biological value of the bioactive compounds in this matrix and their health benefits on the human health.
Suitable methods were applied for the extraction procedure and the analysis of the targets.
However, from the quantitative analytical point of view, the sections 4.3.1, 4.3.2, 4.3.3 and 4.3.4 in the material and method sections explain the quantitative approach used in absolutely superficial way. In particular, Limit of detection, Limit of quantification, repeatability, reproducibility, recovery of the extraction and recovery of the method were not discussed.
All these parameters are necessary when quantitative data are reported. Then I suggest to improve your method for the quantitative purpose. You can refer to the following link for the official procedure from EURACHEM GUIDES:
Another critical point is the quantification of coeluted compounds. The calibration curves chosen to quantify the peaks must be specified.
At least one chromatogram for each class of compounds should be reported, maybe those one of the richest samples.
Too much self-citations are reported.
In the attached file you can find specific corrections to improve the quality of the manuscript.
On the basis of this consideration, I suggest to improve the quantitative method adopted by considering the parameters cited above (LOD, LOQ, recovery, etc…) and with the other suggestions.

Author Response
"Please see the attachment."

Reviewer 3 Report
The manuscript Phytochemical characterization of Habanero pepper's by-products grown in two different soil types from Yucatán, México, provides useful information and be published after a critical English language check.
Author Response
"Please see the attachment."

Round 2
Reviewer 2 Report
All the suggestions were taken into account and the quality of the manuscript has been improved, especially by adding data concerning the method validation.
I recommend the manuscript for publication.